# The internal and external factors influencing talent development of athletic talent among Saudi Arabia's twice exceptional elite athletes: A comprehensive study

**Abdulhamid A. Alarfaj**[1], **Marwa M. Hassan**[2], **Refah M. Aljohar**[1], **Fahad A. Almuaili**[1], **Mohamed D. Hassan**[2]*

1 Department of Special Education, College of Education, King Faisal University, Al-Ahsa, Saudi Arabia,
2 Department of Physical Education, College of Education, King Faisal University, Al-Ahsa, Saudi Arabia

* mdhassan@kfu.edu.sa

## Abstract

This study investigated the internal and environmental factors contributing to talent development among twice-exceptional elite athletes. Data were gathered through in-depth interviews with athletes diagnosed with a disability who achieved notable sports accomplishments at local, regional, or international levels. The sample included 21 athletes aged 18–56 years. Analysis revealed ten key factors that shaped the personalities of elite athletes, fostering exceptional performance. These factors were categorized into five internal components and five environmental motivations. Additionally, the study identified optimal timing for leveraging motivational factors to sustain talent development. The findings were discussed in relation to the Mega Model hypotheses, highlighting the alignment of these factors with success indicators among twice-exceptional athletes.

## Introduction

The notion of talent development has garnered significant attention recently due to its profound impact on fostering self-concept and maximizing individual potential [1–3]. Moreover, talent development plays a pivotal role in cultivating innovative thinking and creativity, thereby enhancing future competitiveness [4–6], developing communities [7], and closing achievement gaps, especially among underrepresented gifted students [8]. The intensifying competitive dynamics in political and economic realms across countries, particularly in human resources development, have underscored a heightened focus on talent development across diverse fields [9–11].

Although the twice-exceptional are one of the gifted categories, they fall within the most prominent categories that do not have sufficient knowledge about the most appropriate ways to enhance and develop their talents in various fields [12–14]. Amran & Majid (2019) conducted a systematic review on (44) experimental studies to evaluate and analyses interventions and learning practices among individuals of the twice-exceptional [15]. The results indicated

**Data Availability Statement:** Due to ethical restrictions and to protect the privacy of the

participants, particularly individuals with disabilities, the full transcription of interview data cannot be made publicly available. Participants were informed during the interviews and file reviews that their data would remain confidential and not be disclosed publicly. The minimal data required to replicate the study's results and verify the qualitative analysis has been provided in Table 1. For further inquiries or access to specific data, please contact the Data Ethics Committee at +966135899773 or via email at ialjreesh@kfu.edu.sa.

**Funding:** The authors extend their appreciation to the King Salman center For Disability Research for funding this work through Research Group no KSRG-2022-090. The funders had no role in study design, data collection and analysis, decision to publish, or preparation of the manuscript.

**Competing interests:** The authors have declared that no competing interests exist

that some forms of recommended interventions were unsuitable for generalization, and they lack comprehensiveness, as they neglect the psychological, social, and personal aspects [15].

Over the past decades, research efforts have aimed to advance our understanding of suitable methodologies for nurturing the diverse talents of twice-exceptional individuals. However, existing empirical studies may not fully support this endeavor due to their narrow focus on specific aspects, neglecting the broader contextual considerations [13].

Arnstein (2022) posits that effectively identifying and providing appropriate developmental services for twice-exceptional individuals necessitates integration across all facets of the school environment, including parents and specialists [14].

To achieve a profound understanding of talent development, a methodological approach involves conducting analytical studies on individuals, tracking developmental changes influenced by internal and external factors [16], tailored to the diverse requirements and capacities specific to each field. This approach offers a comprehensive framework for elucidating the reasons behind ineffective interventions and the challenges faced by twice-exceptional individuals in realizing their full potential within educational settings [17, 18]. Furthermore, this approach also provides a rationale for the success observed in extracurricular interventions across non-academic domains, including arts, dance, and sports [19].

For instance, specialized transition programs in elite sports tailored to specific fields [20], exemplify this approach, such as those seen in the Paralympics catering to twice-exceptional athletes [21]. Previous studies have predominantly emphasized the impact of sports training on talent development, overlooking the significance of various other factors that could also influence this process. These factors include familial, economic, and educational backgrounds, as well as age, developmental stage, and specific condition characteristics, among others [22–25]. The increasing emphasis on training and skill development warrants examination within a comprehensive framework that considers the interplay of these diverse influences on talent development [26–28]. For instance, the ability to effectively balance athletic demands with educational commitments is particularly crucial for elite athletes, given its profound impact on performance [22, 29–31]. The advancement of elite twice-exceptional athletes is hindered by challenges in accessing appropriate training and development opportunities, a situation that is compounded by difficulties in accurately identifying and recognizing their twice-exceptional talents [32–34]. These obstacles are influenced not only by the unique characteristics of these athletes but also by broader societal and environmental factors [21, 35, 36]. In addition to the aforementioned considerations, other factors such as social influences from friends and parents, as well as demographic variables, may also impact the development of performance and excellence levels among twice-exceptional athletes [35–37]. This prompts a broader examination of variables that could potentially influence the developmental trajectory and performance outcomes of elite twice-exceptional athletes [34, 35, 38].

This study investigates the internal and external factors influencing the development of athletic talent among elite twice-exceptional athletes. Elite twice-exceptional athletes are individuals with one or more disabilities, diagnosed according to Saudi Arabia's disability classification system and the World Health Organization's guidelines. They exhibit exceptional abilities in Paralympic sports and have attained a minimum of three local and international accolades.

These research questions were prompted by the authors' inquiry into:

- What internal factors act as catalysts for the development of athletic performance levels among elite, twice-exceptional athletes?

- Which environmental factors contribute to stimulating athletic performance development among elite twice-exceptional athletes?

## Materials and methods

We employed a qualitative approach rooted in phenomenological research, which aligned well with the study's objectives [39]. According to Creswell and Poth [40]. This approach is adept at uncovering the essence of a phenomenon or the human experience encountered across different life stages [41, 42]. It facilitates the identification of factors and underlying influences that contribute to the phenomenon, fostering a profound understanding of the issue [43]. Van Manen (2014) noted that practicing the methodological exploration of phenomenology had to put the phenomenon in its natural context to specify and organize the various givens that made understanding the concepts more realistic and comprehensive. Thus, it creates a clear and exact methodological framework for the phenomenon, determining the basics of sports success to build a system. This framework resulted from the real experiences of the students, sponsoring the disabled promising champions [44].

### In depth interview model

Drawing from the qualitative literature, we identified a specific set of tools for data collection relevant to conducting in-depth interviews for our phenomenological research [40, 42]. As noted by Moser and Korstjens, face-to-face in-depth interviews are the most propitious for phenomenological research methodology. Similarly, Creswell and Poth emphasized this method [40, 45]. Many studies and literature have recommended it posing general and open questions and then focusing more on collecting the data that explain the basics and essential framework of the experience being examined to finally come to a better understanding of the factors common to individuals' experiences [40, 42, 43, 46, 47].

Subsequently, we designed an in-depth interview model comprising 94 items organized into nine fundamental axes: personal data, social data, economic data, health development history, personal and behavioral aspects, academic growth history, interests, hobbies, and contributions of caregivers. This tool underwent rigorous review by four professors specializing in special education, sports education, and psychology. Following their feedback, several items were modified, replaced, or removed, resulting in a final version containing 88 items distributed across the same nine axes, providing comprehensive coverage of developmental history across various domains.

### Portfolio analysis

To achieve the study's objectives, we designed a model to analyze the portfolio of the athletes participating in the current study. The model was important in specifying the study sample and deciding whether to accept or exclude participants. The tool was designed in collaboration with a professor who specialized in physical training at King Faisal University. It comprised 88 items with 7 major axes that included physiological indicators, physical fitness variables, anatomical characteristics, sports achievements, nomination and training, psychological qualities and behavioral traits, and the level of genetic implications in the family. The tool was then sent to 4 professors who specialized in special education and psychology; it was applied to some athletes of the disabled clubs in Al-Ahsa City. The preliminary examination of the data indicated that a significant number of changes needed to be made, such as the total removal of three axes (physiological indicators, physical fitness variables, and anatomical characteristics) due to a dearth of information regarding them in the private clubs or sports unions. There were still four axes, totaling 54 items in their final configuration.

## Participants selection and criteria

Participants for this study were selected using purposive sampling to ensure they met specific criteria aligned with the study's objectives [45]. Direct communication was established with individuals from the Ministry of Sport in the Kingdom of Saudi Arabia (KSA) and administrators of clubs for the disabled to identify and engage the targeted sample. Selection criteria included:

**Disability Diagnosis:** Participants were required to have a documented disability according to KSA disability regulations and WHO classification.

**National Achievements:** Each participant had achieved a minimum of three national-level accomplishments.

**Regional or International Achievements:** Participants also demonstrated at least one achievement at the regional or international level.

**Voluntary Participation:** Participants or their guardians voluntarily chose to participate in the study.

Table 1 highlights the distribution of volunteer participants in the study based on sport type, disability type, age, and disability onset. The key findings from Table 1 can be summarized as follows:

Participant Diversity (Age and Disability Type):

Table 1 reveals significant diversity in age groups, with participants ranging from 19 years old (the youngest) to 56 years old (the oldest).

**Table 1. Participant distribution by sport type, disability type, age, and timing of disability.**

| Name | Age | Type of Disability | Time of Disability | Type of Sport |
|---|---|---|---|---|
| P1 | 32 | Hemiplegia | Accident in 2012 at age 22 | Table Tennis (TT4) |
| P2 | 19 | Lower Hemiplegia | Accident at age two | Shot Put (F53) |
| P3 | 24 | Cerebral Palsy | During birth / Lack of oxygen | Short-distance running (100m, 200m, 400m, T37) |
| P4 | 22 | Left leg disability | Medical error during vaccination at age 5 | Discus and Shot Put (F57) |
| P5 | 19 | Foot disability | Spinal tumors at age two | Wheelchair Racing (100m, 200m, 400m, T34) |
| P6 | 45 | Left limb disability | Medical error causing polio at age 8 | Table Tennis (TT9) |
| P7 | 26 | Lower limb disability | Polio (Hemiplegia) | Wheelchair Racing (100m, 200m, 400m, 800m, T35) |
| P8 | 40 | Amputation of foot | Work injury | Weightlifting (88 kg) |
| P9 | 19 | Quadriplegia | Disability since birth | Boccia (BC3) |
| P10 | 19 | Weakness in limb muscles | At age 4, muscle weakness developed | Wheelchair racing (T54 classification) |
| P11 | 52 | Right leg amputation due to diabetes | At age 50, after amputation | Taekwondo (former), Archery (current) |
| P12 | 36 | Paralysis in lower limbs | At age 17, due to a car accident | Wheelchair basketball |
| P13 | 19 | Amputation of foot due to cancer | At age 16 | Wheelchair racing (T42), Muay Thai (limb category) |
| P14 | 46 | Poliomyelitis | Since birth | Powerlifting (72 kg category) |
| P15 | 56 | Poliomyelitis due to medical error | Since birth | Powerlifting (75 kg category) |
| P16 | 37 | Rickets | Since birth | Powerlifting (107 kg category), Shot put (F37) |
| P17 | 45 | Right leg amputation | At age 17 | Powerlifting (100 kg category) |
| P18 | 52 | Poliomyelitis | Since birth | Powerlifting (100 kg category) |
| P19 | 23 | Autism spectrum disorder | Since birth | Ice skating (25m, 55m, speed skating) |
| P20 | 36 | Intellectual disability | Since birth | Running, marathon, ice running, cycling, bowling, athletics (T20 classification) |
| P21 | 37 | Hemiplegia | Fall at age 1.5 | Discus throw, shot put (T34 classification) |

There is also notable variation in disability types, including physical impairments such as paraplegia and limb amputations, as well as intellectual disabilities and autism spectrum disorders.

Disability Onset and Causes:

Disabilities are categorized into those present since birth or early childhood (e.g., polio or oxygen deprivation during birth) and those acquired later due to accidents or medical errors (e.g., injuries caused by accidents or vaccination complications).

The timing of disability onset reflects its psychological and social impact, with some participants acquiring their disabilities during critical life stages, such as adolescence or adulthood.

Sport Type:

Participants engage in a wide range of sports that require diverse physical abilities:

Individual sports: such as powerlifting, track and field (e.g., shot put, wheelchair racing).

Team sports: such as wheelchair basketball.

Para-specific competitive sports: including Boccia and classification-based sports (e.g., T and F categories).

This diversity demonstrates the opportunities available for individuals with disabilities to participate in various sports, taking into account the specific classifications associated with each disability type.

Table 2 shows that participants' achievements primarily included national medals within Saudi Arabia and the Gulf States, with 80% of them achieving achievements at the continental and international levels. Notably, some participants were distinguished with Olympic medals in discus throw and ice-skating.

This structured approach ensured the study included a diverse and accomplished group of athletes with disabilities, reflecting a broad spectrum of sporting disciplines and achievements.

## Data analysis and validity

The interviews were conducted between October 19, 2022, and January 9, 2023. Two members of the research team, in collaboration with an independent athlete with extensive experience in disability sports and an executive manager of a disability sports club in the Kingdom, conducted 12 interviews, representing 57% of the total interviews. Additionally, 7 face-to-face interviews, comprising 33% of the total, were conducted by a research team member and the independent athlete. To ensure the participation of female athletes, the researcher personally conducted 2 interviews, which accounted for approximately 10% of the total., resulting in approximately 26 hours of recorded material totaling 1552 minutes. The collected dataset included a diverse range of content, comprising 359 megabits of images, documents, and participants' achievements.

The data analysis proceeded through several systematic stages:

**Transcription and documentation.**   Utilizing Transcriptor software, interview data from Zoom sessions were transcribed, resulting in 282 pages of text containing 107,167 words extracted from 14-recorded interviews. Additionally, original interview data from face-to-face sessions were captured directly from participants' statements.

**Triangulation approach.**   Employing a triangulation method, the dataset was divided among three researchers. Two researchers independently analyzed the data of ten and eleven participants, respectively.

**Content-directed exploratory analysis.**   The analysis employed an exploratory approach focusing on open coding and thematic development derived directly from the data. Both probability and nonprobability sampling techniques were utilized to enrich the analytical process.

**Peer review and validation.**   Results underwent a peer review process between the two researchers, following the framework proposed by Johnson and Christensen (2019). This

**Table 2. Participants descriptive.**

| Characteristic | Frequency | Percent | Descriptive St. |
|---|---|---|---|
| **Age Groups** | | | |
| 10–19 | 5 | 23.8% | $N = 21$ |
| 20–29 | 4 | 19% | $\bar{x} = 32.59$ |
| 30–39 | 5 | 23.8 | $\sigma = 14.01$ |
| 40–49 | 4 | 19% | Skewness = 0.132 |
| 50–59 | 3 | 14.3% | |
| **SEX** | | | |
| Male | 19 | 90.5% | |
| Female | 2 | 9.5% | |
| **Disability** | | | |
| **Characteristic** | **Frequency** | **Percent** | **Descriptive St.** |
| Impaired mobility | 19 | 90.5% | |
| Developmental disability | 2 | 9.5% | |
| **Representativeness in KSA** | | | |
| Eastern Region | 9 | 42.9% | |
| Middle Region | 7 | 33.3% | |
| Western Region | 3 | 14.3% | |
| Northern Region | 1 | 4.8% | |
| Southern Region | 1 | 4.8% | |
| **Para-Olympic Sports** | | | |
| Weightlifting | 6 | 22% | |
| Wheelchair basketball | 2 | 7.4% | |
| Archery | 1 | 3.7% | |
| Wheelchair racing | 5 | 18.5% | |
| Ice skating | 2 | 7.4% | |
| Discus throw | 2 | 7.4% | |
| Cycling | 1 | 3.7% | |
| Table tennis | 2 | 7.4% | |
| Shot put | 4 | 14.8% | |
| Boccia | 2 | 7.4% | |

iterative review aimed to validate the analysis, refine major themes, and identify significant subthemes within the dataset [48].

The following passage has been added to clarify the credibility of the interviews and qualitative data analysis:

Data from the participants were collected using the study tool through in-depth interviews with the athletes or their guardians, depending on the individual's situation. Two members of the research team (Fahad and Rifa) conducted the interviews, along with a former athlete with a disability who is also the Executive Director of a disability sports club. This individual holds a Master's degree in Special Education and has 28 years of experience in the field. The data analysis process included several stages:

1. The researchers transcribed the interview and focus group recordings using Sonix (https://sonix.ai/), a platform designed for this purpose that supports Arabic and employs artificial intelligence for language and text recognition across contexts.

2. An exploratory content-driven analysis approach was used, which involves open coding and themes derived from the data, applicable to both probabilistic and non-probabilistic samples [49].

3. During the qualitative data analysis, the researchers performed triangulation of raw analysis by reviewing the data analysis process with a peer—a colleague in a gifted education PhD program—and a special education expert with 28 years of practical experience, known as peer review (Johnson & Christensen, 2019), to ensure the credibility of the analysis and review the main and sub-themes [48].

4. Additionally, the researchers presented the final analysis results to an independent researcher for external audit, a method involving external experts to assess the study's quality. The researcher reviewed the primary analysis and the initial review to determine the level of agreement and ensure the quality of identifying the main and sub-themes [48].

**Synthesis of findings.** To ensure consistency and alignment across analyses, the lead author reviewed and synthesized the results from all researchers, identifying and specifying the main themes and subthemes emerging from the study's dataset.

This rigorous and systematic approach to data collection and analysis employed advanced methodologies, including technological tools, collaborative triangulation, and thematic exploration, ensuring robustness, reliability, and validity in the study's findings. The integration of peer review and synthesis further strengthened the analytical process.

## Ethical considerations

This study adhered to rigorous ethical standards, receiving approval from the Research Ethics Committee of King Faisal University (KFU-REC-2022-OCT–ETHICS195). Prior to participation, all individuals provided informed written consent, and strict confidentiality measures were implemented to safeguard participant privacy and confidentiality throughout the study. These ethical safeguards were paramount in ensuring the integrity and ethical conduct of the research, aligning with international standards for research involving human subjects.

## Results

The development of talent requires the presence of both internal and external factors. These factors must work together to ultimately lead to the development of talent. This principle serves as a fundamental cornerstone of the Mega Model for Talent Development [50].

Identifying the influencing factors of Twice-Exceptional Elite Athletes: A Mega Mode Approach. In our investigation, we applied a Mega Mode framework to delineate factors contributing to the identity of twice-exceptional elite athletes. This framework categorizes these factors into internal and external components, both of which are integral to motivating individuals toward achieving excellence.

The internal components encompass general and specialized capabilities within the field, as well as social and psychological skills, and growth experiences [51].

Conversely, the external components are associated with opportunities and favorable conditions related to timing, type of support, and duration [1]. These external influences provide the necessary environment and resources for talent to thrive [1].

## Variability and commonalities among elite athletes with disabilities

Despite the diversity in participants' disabilities and levels of athletic excellence, our study revealed striking similarities in the experiences of elite athletes. This finding aligns with

existing research on twice-exceptional individuals, which highlights consistent general qualities within this group. For instance, studies emphasize the importance of emphasizing strengths in interventions for the twice-exceptional, rather than solely focusing on weaknesses [15, 52, 53].

It is important to note that our study does not seek to generalize results due to limitations in methodology and sample size. Instead, our primary objective is to explore the developmental pathways of elite athlete talents. Therefore, our study deliberately focused on a specific and limited sample, acknowledging disability as a fundamental shared factor among participants' experiences.

By concentrating on these shared experiences within a targeted sample, our study aims to contribute nuanced insights into the unique developmental trajectories of elite athletes with disabilities. This approach underscores the importance of understanding individual strengths and contextual factors in talent development among this exceptional athlete population.

## Themes and factors impacting excellence among twice-exceptional elite athletes

Our analysis of elite athlete experiences among the twice-exceptional revealed ten major themes categorized into five internal components and five environmental factors. Collectively, these themes contributed positively to the excellence demonstrated by twice-exceptional athletes.

Drawing from our findings, we developed a hypothesis that elucidates the interplay between external and internal environmental factors influencing elite athletes' path towards excellence, as illustrated in **Fig 1**. This hypothesis offers valuable insights into the complex dynamics shaping the development and success of elite athletes with unique abilities and challenges.

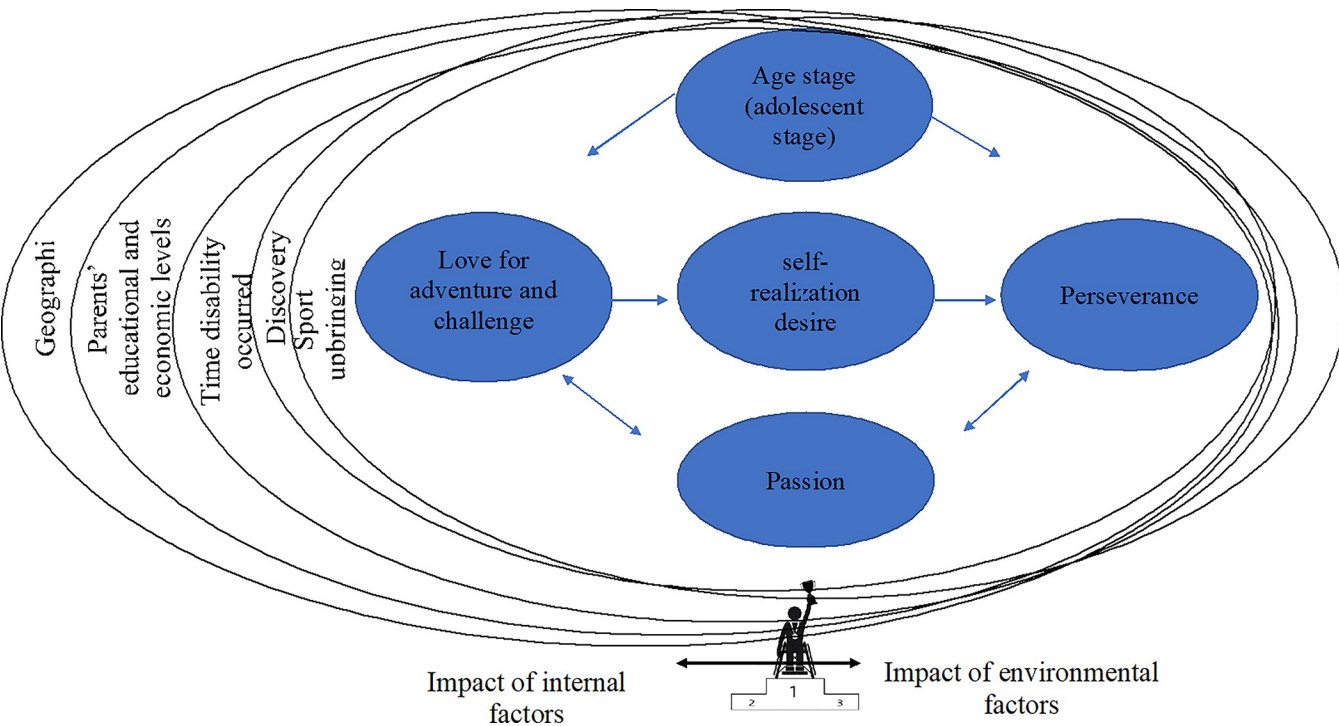

**Fig 1. An overview of the work being done by internal and external components and incentives.**

Through this study, we aim to contribute substantively to the understanding of talent development and achievement within the context of twice-exceptionality in sports, highlighting the multifaceted influences that drive excellence among this exceptional athlete population.

## Internal factors

The internal factors identified in our analysis encompassed several key elements contributing to the success and excellence of elite athletes among the twice-exceptional population.

The results presented in Table 3 highlight the internal factors that contributed to the participants' success in the sporting field. These factors provide insight into the personal motivations, characteristics and developmental stages that shaped their achievements in various sporting disciplines. The following is an explanation of what is stated in Table 3:

*Perseverance*: Participants unanimously attributed their achievements to perseverance and rigorous training regimens. They emphasized that perseverance not only elevated their performance levels but also enhanced their professionalism within their respective sports do-mains. Previous research by Durand-Bush & Salmela (2002) corroborates the role of perseverance and continuous training in advancing sports performance to expert lev-ells [54–56]. Perseverance included daily training hours to which they committed themselves. Training sessions ranged from 3–4 hours daily, sometimes in two sessions: morning and evening.

*Desire for Self-assertion*: Participants emphasized the need for self-assertion to overcome challenges. The deficiency that the twice-exceptional individual felt motivated them to work forcefully to achieve professionalism and achievements as compensation for that deficiency [57]. This incentive triggers internal abilities to build experience [58, 59]. The need for self-satisfaction, objective achievement, and purposeful practice is a means for improving the level of performance and perfection [60–63].—Self-assertion motivated them to constantly improve their sports performance and professionalism.—Achievements helped them prove their potential to others and overcome societal beliefs about their disabilities. Participant (3) mentioned that his first participation in the special sports field was an international one, in which he was a member of the kingdom's team and was awarded gold medals. He said, "The first match I played was a strong start for me; in the second, I felt that all eyes were gazing at me."

**Table 3. Internal factors.**

| Internal Factors | Description |
|---|---|
| Perseverance | • All participants attributed their success to perseverance and constant training.—Perseverance included daily training hours to which they committed themselves.—Training sessions ranged from 3–4 hours daily, sometimes in two sessions: morning and evening. |
| Desire for Self-assertion | • Participants emphasized the need for self-assertion to overcome challenges.—Self-assertion motivated them to constantly improve their sports performance and professionalism.—Achievements helped them prove their potential to others and overcome societal beliefs about their disabilities. |
| Passion | • Passion for sports emerged after participants' first accomplishments in their careers.—International accomplishments created a stronger passion than local achievements.—Passion was a driving force behind performance and perseverance. |
| Love for Adventure and Challenge | • Participants found challenge attractive and enjoyable.—Individual games were particularly appealing due to the increased challenge compared to team sports.—Love for adventure and challenge led to success in various sports fields. |
| Age Stage (Adolescent Stage) | • First achievements were often realized during adolescence.—Positive internal motivation contributed to exceptional achievements during this stage.—Support from outside the school improved self-concept and performance levels.—Adolescence was a crucial period for skill development and transitioning to professionalism in sports. |

Participant (5) justified participation in sports, and the achievements gained overwhelmed him with happiness. Participant (11) mentioned that she wanted to be prominent and go further in the field of sports to prove to others that she could accomplish what ordinary people could not.

***Passion***: For most participants, passion for their sport emerged after their initial accomplishments rather than during childhood. International achievements fueled a stronger passion compared to local successes, indicating the profound impact of recognition on motivation. Research suggests a positive correlation between passion and performance level across cultural backgrounds [62, 64]. Participant 7 expressed a goal-oriented passion, aspiring for global recognition. Similarly, Participant 12 emphasized the critical role of passion in sustaining performance and perseverance, highlighting its significance in elite athletic endeavors.

***Love for Adventure and Challenge***: All participants agreed that challenge was a factor that affected their athletic orientation. The greater the challenge was, the more attractive and enjoyable it would be for them. In fact, the challenge was a trait common to the talented. Many studies have confirmed the active role of positive motivation [65]. Thus, my love for adventure and challenge was reflected in individual athletic games. Participant (2) mentioned that individual games augment challenge because the individual plays individually, unlike collective games with the team where the team gets support from that team." Participant (11) justified her attraction to different fields because of her love for adventure and challenge. Participant (20) revealed that he won gold medals in different fields due to his love for adventure and challenge, through which he got 13 gold medals, regionally and internationally.

***Age Stage (Adolescent Stage)***: Our investigation revealed that the initial accomplishments of Participants occurred during adolescence, with many having minimal prior training, sometimes as short as three weeks. Despite this, they achieved notable success, including gold, silver, and bronze medals, with a significant portion at regional and international levels. This seemingly counterintuitive progression was attributed to a positive internal motivation that outweighed any perceived complications or psychological barriers. Subsequent external support from sports clubs, trainers, and government bodies, such as the Saudi Ministry of Sport, played a pivotal role in improving participants' self-concept and fostering positive perceptions. Notably, enhanced self-perceptions, particularly within the sports domain, correlated with improved performance levels. While some studies indicated a decrease in physical activities during adolescence [63, 66–68], others suggested an increase in bone strength, highlighting the complex interplay of physiological factors [69, 70]. Participants leveraged the opportunities presented during adolescence to excel in their sports, surpassing training stage requirements and achieving professionalism. Remarkable achievements include Participant 15's rapid ascent to international recognition, Participant 6's debut gold medal in an international competition, Participant 16's consistent success in international championships, and Participant 17's groundbreaking performance in weightlifting.

## Environmental factors

The factors that mostly affect participants are the following:

The results summarized in Table 4 explore the environmental factors that influenced the participants' sporting careers. These factors, which include challenges and benefits, highlight the role of external circumstances in shaping their sporting development and achievements. The following is an explanation of what is stated in Table 4:

**Parents' Academic and Economic Levels:** The academic and economic backgrounds of participants' parents varied, encompassing levels ranging from good to poor. While neither directly impacted performance development or sports professionalism, their influence was

**Table 4. Environmental factors.**

| Environmental Factors | Challenges | Benefits |
| --- | --- | --- |
| **Parents' Academic and Economic Levels** | • Low academic and economic levels hindered athletes' ability to balance sports and other life pursuits.<br>• Financial constraints affected access to training equipment and opportunities. | • The advanced academic level contributed to supporting and motivating athletes, especially in the early stages of discovery and athletic preparation.<br>• A good financial situation was a motivating factor in accessing higher-quality services and athletic preparation.<br>• Good academic and economic levels indirectly impacted performance development by encouraging athletes to pursue a balanced life |
| **Geographical Location** | • Participants in areas far from major cities found it difficult for athletes to access scouting and training services.<br>• Participants often moved to cities with better resources to pursue their athletic careers. | • Geographical location significantly impacted access to sports infrastructure and support services.<br>• Major cities with better facilities and support systems facilitated athletic success. |
| **Time Disability Occurred** | • Timing and cause of disability influenced participants' ability to adapt and excel in sports.<br>• Late-onset disabilities posed psychological challenges that hindered participation in sports activities. | • Early onset disabilities were easier to adapt to compared to disabilities acquired later in life.<br>• Early diagnosis of disability was one of the most distinguished characteristics of athletes . |
| **Discovery** | • The lack of qualified teachers in special education centers and programs reduces the chances of discovering sports talents.<br><br>• When the discovery of an athlete's talent is delayed, this reduces the level of sports achievement that is achieved based on the participants' experiences.<br>• The presence of the participant in a family/school environment that is not supportive of sports activities reduces the chances of discovering his talent. | • Sports clubs for people with disabilities and medical rehabilitation centers were the most important source of athletes to discover athletes with motor disabilities, while schools were more important for those with developmental disabilities.<br>• Family/school support was also crucial in discovering and developing athletic talent.<br>• Many participants were discovered before they realized their own talents.<br>• Starting sports early, especially during middle school, was important for success |
| **Intensive Professional Athletic Upbringing** | • Sports clubs do not provide professional training for athletes due to limited resources.<br>• Some sports clubs lack adequately qualified coaches to develop athletes' skills to achieve high-performance levels.<br>• The duration and intensity of training sessions offered in clubs for persons with disabilities do not meet the necessary requirements to reach the appropriate professional level. | • Professional training camps with qualified coaches, intensive training plans, and adequate equipment were essential for athletic development.<br>• Qualified coaches played a vital role in developing and motivating athletes.<br>• Intensive training sessions prior to competitions were considered a high-priority factor for success from the athletes' perspective.<br>• The availability of necessary equipment and resources enhanced training experiences and performance outcomes. |

indirect. Participants from families with higher academic and economic standings were motivated to juxtapose sports excellence with pursuing conventional life goals such as higher education, employment, health maintenance, and marriage. Conversely, participants from lower academic and economic backgrounds faced challenges in balancing sports commitments with other aspects of life. Many were unable to pursue further education beyond secondary schooling, secure desirable employment opportunities, or establish stable marital relationships. These challenges were reflected in their sports performance, as financial constraints hindered access to specialized equipment and training facilities, exacerbating the impact of their socioeconomic status on athletic endeavors. Moreover, inadequate parental caution and knowledge potentially contributed to the occurrence of disabilities. Previous studies have underscored the influence of parental educational levels on adolescents' achievement motivation and athletic performance [71, 72]. These findings offer insights into the disparities among twice-exceptional participants and underscore the crucial role of socioeconomic factors in shaping athletic outcomes.

*Geographical Site*: The geographical location during both the developmental and postpubertal stages significantly influenced the professionalism of participants. That sounds logical, while some big cities enjoyed development, prosperity, and many services, some small cities and villages suffered from a shortage of services, especially for the twice-exceptional ones in the sports domain.

The foremost needs associated with the geographical site of the city are outlined in the following: sports clubs for training and qualification, sports needs and tools, special equipment that matches the type of disability competition, wheelchairs and helmets for Boccia sport, financial support, sports arenas, competent coaches, and enlightenment through activities. These factors help raise the number of twice-exceptional champions in some cities where they won gold medals in international competitions. We found that because of these qualities, a certain phenomenon spread among participants; they moved to such cities depending that the situation was suitable, as 7 of them did. For them, the move to cities shortens time and effort for accomplishments as long as potentials and privileges exist. The comparison we made between large and small cities revealed a great difference in the performance of the twice- exceptional individuals. Participant (12), for example, mentioned that the club he belonged to rated first in the number of medals it got, which was a quantum leap from the researchers' viewpoint.

Many participants declared that the site helped make them either distinguished in the field or not. Nine of them attributed their success to life in larger cities where excellent clubs, great potential, and unlimited support were available. Participant (11) mentioned that she could not imagine her future if she lived in a small city. She attributed her success to the distinguished city where she lived. In contrast, participant (1) pointed out that "the city didn't provide support to those with special needs when the disability occurred".

*Time Disability Occurred*: The participants were divided into two groups based on the timing and cause of their disability: those born with the disability and those who acquired it later in life, often due to car accidents. This classification revealed significant differences in the ease of adapting to the disability and its impact on performance development. Individuals who acquired disabilities later in life faced greater challenges in adaptation. Participant (7) mentioned that they live naturally with their disability, not considering it an issue, and engage in all life activities normally. Participant (21), who acquired their disability in early childhood, adapted to it and, from a very young age, directed their focus towards sports and developing their talent. In contrast, Participant (8), who became disabled after reaching adulthood, needed some time to adjust to the disability and relied heavily on peers to help them through this period and encourage the development of their talent. Similarly, Participant (1), who also acquired a disability after adulthood, required the presence of peers and interaction with individuals with the same disability to adapt to their condition and motivate them to develop their talent.

Many struggled to accept their new circumstances, which hindered their participation in sports activities. This finding is consistent with research by Boyce and Wood (2011), indicating that adapting to a disability is easier when it occurs earlier in life. In contrast, those who adapted easily achieved remarkable accomplishments and competed at various levels, balancing their athletic pursuits with other aspects of life [73]. Gignac et al. (2000) observed that individuals with chronic disabilities often develop effective coping strategies, leading to improved performance and independence [74]. These insights highlight the importance of early adaptation to disability and the role of active compensatory strategies in achieving success in sports and maintaining a balanced life.

*Discovery*: This factor holds significant importance in talent development, serving as the primary gateway to nurturing talent. Our investigation delved into the participants' backgrounds to identify any common elements in their discovery process. Remarkably, 12 of them were discovered before their talents were recognized. Many were identified as club members scouted for twice-exceptional talent during indoor contests and matches. Others were unearthed through physical therapy, while five were identified through familial or school channels. Despite their delayed entry into clubs and competitions, the majority commenced their journey after the secondary stage, underscoring the significance of early engagement, particularly in the realm of sports.

*Intensive Professional Athletic Upbringing*: Undoubtedly, professional training significantly influences the development of exceptional achievement, a notion supported by educational literature [75]. Authors have also attributed performance level disparities between genders to variations in training intensity and continuity, as well as coaches' expertise [50]. All participants who garnered local, regional, or international medals underscored the pivotal role of training resources in enhancing athletic competence and attaining remarkable feats. According to their accounts, these training camps were distinguished by: (A) highly qualified professional coaching; (B) intensive training regimes tailored for competition readiness; and (C) the provision of necessary training equipment. Participants unanimously acknowledged the indispensable role of professional coaches in their journey to excellence. These coaches were adept not only in the fundamental tactics of athletic games but also in motivating players to overcome obstacles. Participant (1) attributed his excellence to his professional coach, while Participant (2) highlighted the intensive training regimen he underwent prior to the West Asia Championship, resulting in a second-place ranking after just 15 days of focused training. Many participants emphasized the rigorous training periods preceding competitions, noting how these camps honed their skills and prepared them for future championships. They described the camps as characterized by seriousness, with daily schedules and specialized dietary plans.

Participant (12) noted how his participation in an outdoor camp transformed his understanding of the game, describing himself as fortunate for the experience. Additionally, most participants emphasized the importance of having access to necessary tools, equipment, and human resources, which were often lacking in regular training camps. Participant (15) highlighted the comprehensive provision of training equipment, including medical and communication devices, at these camps.

## Discussion

Numerous studies in the literature have affirmed the potential for intentional development across various domains [76]. However, a pertinent question arises: What were the key factors responsible for nurturing sports talent among elite, twice-exceptional athletes?.

To elucidate these capacities and comprehend their operational mechanisms, we scrutinized the experiences of individuals who had attained the benchmarks indicative of talent. Accordingly, we investigated both internal factors and environmental factors, discerning how they catalyzed talent development. Given the diverse experiences of the participants within the twice-exceptional category, we conducted comparative analyses to benchmark their actions' efficacy and their influence on developmental trajectories under varying circumstances, contrasting ideal conditions with those lacking such support.

The participants' experiences unveiled a synergistic interplay among internal factors and other components, alongside interactions with environmental factors, both individually and collectively. Furthermore, a positive and catalytic interaction was observed between internal and environmental factors. Each factor stimulated the emergence and evolution of the other, contingent upon the individual's age and developmental stage within the talent trajectory.

### The role of internal factors in developing sports performance

The transition into the realm of sports followed a similar trajectory for participants, yet the nuanced interplay of various capabilities, representing the intended practice method, differed significantly. This variance influenced the breadth and depth of skill development achievement. For many participants, adolescence marked the on-set of their sports journey,

coinciding with the commencement of diverse sporting pursuits [51]. These endeavors often catalyzed emotional qualities such as a love for challenge and self-assertion [76].

Motivation serves as an internal engine for mastering performance and improving psychological, social, and life skills [77]. Previous research emphasizes the significance of internal emotional components such as passion, satisfaction, desire, and self-assertion in enhancing athletic performance within structured practice environments [63]. Passion and strong emotions, sometimes referred to as "hallucinations," have varied effects on athletes. They not only contribute to feelings of achievement and vitality during moments of victory but also catalyze athletes to set objectives and adopt individual methodologies, ultimately leading to excellence [64].

Studies centered on the hypotheses underlying the model and the requirements of the twice-exceptional population emphasized the importance of leveraging strengths, addressing weaknesses, implementing credible interventions, and tailoring approaches to individual needs [78–81]. These efforts were aimed at fostering supportive internal psychological traits, such as passion, perseverance, and assertiveness, which were crucial for talent development.

Notably, many individuals struggled to accept their new circumstances, which hindered their participation in sports activities. This finding is consistent with research by Boyce and Wood (2011), which suggests that adapting to a disability is easier when it occurs earlier in life. In contrast, those who adapted successfully achieved remarkable accomplishments and competed at various levels, balancing their athletic pursuits with other aspects of life [82]. Moreover, Gignac et al. (2000) observed that individuals with chronic disabilities often develop effective coping mechanisms, further emphasizing the importance of psychological resilience in overcoming challenges [73]. These insights highlight the importance of early adaptation to disability and the role of active compensatory strategies in achieving success in sports and maintaining a balanced life.

Motivation surged when favorable environmental factors were aligned with internal traits, enabling disabled individuals to achieve remarkable performance commensurate with their age and athletic trajectory. These internal components served as catalysts for overcoming challenges and sustaining motivation and growth.

Our findings indicate that elite athletes navigate this transition from capability to competence more easily than from competence to experience. Internal components play a crucial role in this process, enabling athletes to enhance their performance and surpass competitors. While environmental factors like academic and economic levels, as well as geographical location, have less influence, internal motivation for achievement and self-assertion emerges as key drivers, often described as falling in love [19].

## The role of external factors in developing sports performance

Disparities in parental education and economic status, along with opportunities for talent discovery linked to geographical location, significantly influenced the development of emotional qualities essential for nurturing sports talent. Champions who benefited from supportive environments and optimal interactions achieved exceptional performance levels, earning numerous international gold medals and setting records. In contrast, those lacking such nurturing opportunities did not attain comparable levels of skill and accomplishment. While interaction dynamics played a crucial role, the presence of abilities and the timing of their development also impacted outcomes. Geographical suitability, coupled with parental support and educational advantages in early life, facilitated the athletic development of twice-exceptional individuals, enhancing their chances of achieving high levels of success. According to the "Mega Model," timely opportunities were critical for sustained growth, as each domain required specific timing for optimal development [51]. It is evident that capabilities, when coupled with a

systematic arrangement of quality, size, and timing, facilitate a controlled interaction process with a positive impact on skill development and achievement levels.

The model suggests that capabilities undergo a developmental transition toward competence, leading individuals to become experienced and eventually distinguished in their field [83].

This stage demanded that elite athletes possess psychological and social skills essential for navigating challenges, asserting themselves, confronting obstacles, and outperforming competitors. These skills are often cultivated through immersion in the field and gaining experience through sports practices, as evidenced by elite athletes' experiences in the Olympic Games for the disabled. Research suggests that individuals with more experience tend to develop psychological and social skills such as leadership and communication [84]. Additionally, elite athletes needed to master the intricacies of professionalism and performance arts, underscoring the importance of having expert coaches with a high level of professionalism to facilitate seamless performance and successful navigation of challenges [85]. Participants highlighted the significance of boot camps and the role of professional trainers in accelerating their transition towards excellence and expertise in the field.

We observed the divergence between internal and external factors across various stages of development and noted how the timing of these components influenced talent development trajectories, particularly in the case of twice-exceptional individuals. Timing emerged as a fundamental aspect of the Mega Model theory, wherein the presence of components at opportune moments significantly impacted talent development outcomes. Additionally, each domain had unique temporal needs, highlighting the importance of considering timing in talent development strategies.

## Conclusions

The comprehensive analysis of the data has significantly enhanced our comprehension of the multifaceted factors influencing the development of talent among twice-exceptional elite athletes. These factors operate in diverse ways, with some serving as internal motivators while others function as external influences, shaping talent development trajectories across different age periods. Key internal motivators highlighted by participants include perseverance, passion, love for challenge, developmental stage, and self-assertion. On the other hand, external factors such as parental education and economic status, geographical location, timing of disability onset, discovery processes, and professional athletic training are considered environmental motivators.

While not all factors necessarily contribute positively to talent development, the interaction between internal and external factors can transform obstacles into constructive challenges that foster talent growth. These findings offer valuable insights into understanding the complexities surrounding the loss of sports talent among the twice-exceptional population and the implications thereof. Moreover, they underscore the pivotal roles played by family, peers, governing bodies, and decision-makers in facilitating talent development at various stages. This holistic understanding provides a roadmap for stakeholders to better support the development of elite athletes with di-verse abilities and challenges.

## Author Contributions

**Conceptualization:** Abdulhamid A. Alarfaj, Mohamed D. Hassan.

**Data curation:** Marwa M. Hassan, Refah M. Aljohar, Fahad A. Almuaili, Mohamed D. Hassan.

**Formal analysis:** Marwa M. Hassan, Refah M. Aljohar, Fahad A. Almuaili, Mohamed D. Hassan.

**Funding acquisition:** Abdulhamid A. Alarfaj.

**Investigation:** Abdulhamid A. Alarfaj, Marwa M. Hassan, Fahad A. Almuaili, Mohamed D. Hassan.

**Methodology:** Abdulhamid A. Alarfaj, Marwa M. Hassan, Refah M. Aljohar, Fahad A. Almuaili, Mohamed D. Hassan.

**Project administration:** Abdulhamid A. Alarfaj.

**Resources:** Abdulhamid A. Alarfaj, Refah M. Aljohar, Fahad A. Almuaili.

**Software:** Marwa M. Hassan, Refah M. Aljohar, Fahad A. Almuaili.

**Supervision:** Mohamed D. Hassan.

**Validation:** Refah M. Aljohar, Fahad A. Almuaili, Mohamed D. Hassan.

**Visualization:** Refah M. Aljohar, Fahad A. Almuaili, Mohamed D. Hassan.

**Writing – original draft:** Abdulhamid A. Alarfaj, Mohamed D. Hassan.

**Writing – review & editing:** Mohamed D. Hassan.

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
