## [Decision Letter · Decision Letter 0]

24 Jul 2024

PONE-D-24-17887Factors Influencing Talent Development and Athletic Performance Among Saudi Arabia's Twice-Exceptional Elite Athletes: A Comprehensive StudyPLOS ONE

Dear Dr. Hassan,

Thank you for submitting your manuscript to PLOS ONE. After careful consideration, we feel that it has merit but does not fully meet PLOS ONE’s publication criteria as it currently stands. Therefore, we invite you to submit a revised version of the manuscript that addresses the points raised during the review process.

ACADEMIC EDITOR

The reviewers acknowledged the potential contribution that this manuscript could make to the existing literature. Nevertheless, the reviewers expressed their concerns about the quality of this manuscript in its current state. The manuscript needs to be developed further before it can be considered publishable. It is encouraged that the author(s) take the reviewers' comments into consideration and revise this manuscript.

If applicable, we recommend that you deposit your laboratory protocols in protocols to enhance the reproducibility of your results. Protocols assigns your protocol its own identifier (DOI) so that it can be cited independently in the future. For instructions see: https://journals.plos.org/plosone/s/submission-guidelines#loc-laboratory-protocols. Additionally, PLOS ONE offers an option for publishing peer-reviewed Lab Protocol articles, which describe protocols hosted on protocols. Read more information on sharing protocols at https://plos.org/protocols?utm_medium=editorial-email&utm_source=authorletters&utm_campaign=protocols.

We look forward to receiving your revised manuscript.

Kind regards,

Tien-Chin Tan, Ph.D.

Academic Editor

PLOS ONE

3. Thank you for stating the following in the Acknowledgments Section of your manuscript: "In The authors extend their appreciation to the King Salman center For Disability Research for funding this work through Research Group no KSRG-2022-090."

Please remove any funding-related text from the manuscript and let us know how you would like to update your Funding Statement. Currently, your Funding Statement reads as follows: “The authors received no specific funding for this work.”

4. We note that your Data Availability Statement is currently as follows: "All relevant data is within the manuscript and its supporting information files."

**Comments to the Author**

1. Is the manuscript technically sound, and do the data support the conclusions?

Reviewer #1: Partly

Reviewer #2: Partly

2. Has the statistical analysis been performed appropriately and rigorously? 

Reviewer #1: N/A

Reviewer #2: N/A

3. Have the authors made all data underlying the findings in their manuscript fully available?

Reviewer #1: Yes

Reviewer #2: Yes

4. Is the manuscript presented in an intelligible fashion and written in standard English?

Reviewer #1: No

Reviewer #2: Yes

5. Review Comments to the Author

Reviewer #1: It is academically and practically beneficial to explore the developmental elements of internal and external influences on this group of twice-exceptional elite athletes. However, there are still several significant issues to consider and clarify. The title, purpose, and content of the article should be consistent, relevant literature should be augmented, the framework for the analysis should be clarified, the interviewees and data collection should be clarified, and the findings and discussion should be reconsidered. These issues have resulted in the current study's conclusions being only partially supported. Additionally, the entire article has a number of formal problems that need to be addressed. These issues and problems have had a significant impact on the overall quality of the current study. Please consider the following suggestions before submitting the article to relevant journals.

Introduction

1. In the title of this paper, the authors state that they investigate the factors influencing the development of talent and performance among elite twice-exceptional athletes. This is inconsistent with the research objective, “This study investigates the internal and external factors influencing the development of athletic talent among elite twice-exceptional athletes” as stated in lines 85-86. Moreover, the authors did not present an analysis of the factors influencing elite twice-exceptional athletes' performance. Therefore, the authors need to pay more attention to the study title, purpose, and content consistency.

2. The authors’ review of existing literature on the factors influencing the development of athletes in competitive sports, or on the factors affecting the development of elite twice-exceptional athletes, is not sufficient and comprehensive. The current authors’ review of relevant previous studies discusses factors at the individual and environmental levels. However, previous studies such as De Bosscher et al. (2006) mentioned that the factors affecting athletic talents development and performance can be categorized into micro, meso, and macro levels. Therefore, the authors may need to refer to De Bosscher et al. (2006) and related previous studies to reorganize the review of related studies.

Veerle De Bosscher, Paul De Knop, Maarten Van Bottenburg & Simon Shibli (2006). A Conceptual Framework for Analysing Sports Policy Factors Leading to International Sporting Success, European Sport Management Quarterly, 6:2, 185- 215, DOI: 10.1080/16184740600955087

Materials and Methods

1. The authors’ description of the respondents in Table 1 allows readers to understand the respondents' composition. However, this section should generally include the respondent's code number, interview date, and their characteristics, such as disabilities classification, specialty sports items, and sports performance. By doing so, readers will be able to better comprehend the source of evidence. This in turn should provide a more convincing picture of the findings.

2. In lines 194-196, the authors state that “the integration of peer review and synthesis further strengthened the analytical process, enhancing the credibility and impact of the research outcomes suitable for publication in high-impact international scientific journals”. However, it seems arbitrary for the authors to assert that their research outcomes are suitable for publication in high-impact scientific journals. Researchers should focus on explaining how their research has enhanced data credibility.

3. In lines 163-167, the authors mention that interview data were collected from 21 respondents on 1 January 2023. The authors state that these interviews add up to 26 hours of documentation. However, due to the relatively large number of 21 respondents, this seems difficult to achieve with only one researcher in a day. It may be necessary for the authors to clarify how interview data were collected, e.g., were the interview data collected on the same day by different researchers?

Results

1. In line 206, the authors mention “Identifying the Identity of Twice-Exceptional Elite Athletes: A Mega Mode Approach”, but then go on to say that “This framework categorizes these factors into internal and external components, both of which are integral to motivating individuals toward achieving excellence.” The authors need to clarify whether or not the concept of identity is an external component.

2. In lines 206-239, the authors mention the mega mode approach as the analytical framework for the study. However, the authors did not explain the approach in the article introduction or in the methodology section. It is recommended that the authors include the approach in the literature review. This includes how it has been used in existing studies and how it was used in this study to analyze the findings. As a result, the discussion section will be more able to reflect on and comment on the approach's applicability.

3. The authors mention in lines 225-226 that “Despite the diversity in participants’ disabilities and levels of athletic excellence, our study revealed striking similarities in the experiences of elite athletes”. However, there may be a misunderstanding in the current authors’ generalization of the influencing factors derived from all the respondents. For example, these subjects came from different sports and may be consistent in internal influences. However, they may show a higher degree of variation in external factors. Therefore, a comparative analysis of respondents with varying background conditions should be presented in the findings section. In addition, the research methodology should be categorized according to the different characteristics of respondents.

4. The authors asked the interviewees to describe their experiences of outstanding athletic performance. From this, the authors extracted the influencing factors and attempt to present the relationships between the factors. Due to the small sample size, the authors will have to clarify the relationship of the factors between each other carefully. For example, does “Perseverance” also affect “Desire for Self-assertion”? Therefore, it is suggested that the authors focus on the induction of the influencing factors rather than the relationship between the factors.

Discussion

1. The authors describe, in lines 415-418, “Given the diverse experiences of the participants within the twice-exceptional category, we conducted comparative analyses to benchmark their actions’ efficacy and their influence on developmental trajectories under varying circumstances, contrasting ideal conditions with those lacking such support”. However, the results of the comparative analyses were not found in the study findings. This section also responds to the question in the findings section whether comparative analyses of data from respondents with different background conditions should be presented first. It is therefore suggested that the authors should revise their discussion to be more in line with the study findings.

2. The current discussion by the authors takes place at the same time when the research findings are in dialogue with previous literature or theoretical applications, e.g. lines 425-444. Apart from the fact that readers may get lost easier in a large amount of text, the discussion of the research findings and previous related studies seems incomplete. Therefore, it is suggested that the authors move the discussion of the findings of internal and external factors in the Results section, such as the similar content in lines 367-375, to the Discussion section for a more comprehensive discussion, as well as discussing them according to the themes, which may make it easier for readers to understand the themes that the authors would like to focus on.

Problems with Article Format

1. What is the keyword 6.?

2. Missing spelling, e.g., line 68 "familial"; line 376 "his factor"; Table 2 title “Environmental Facto”.

3. Names and dates of people appear in internal citations, e.g., lines 211-214 and 215-217.

4. Why do the external components in line 215 appear in bold, but the internal components in line 211 do not?

5. Lines 248-249 repeat the phrase "with unique abilities and challenges”.

6. The font size is inconsistent in many places throughout the text, e.g., in the introduction and lines 152-153, Table 1, etc. 7. The age group in Table 1 contradicts the font size.

7. In Table 1, there should be no brackets around the age group.

8. In Table 1, there should be no need to add punctuation marks for the sports items above Shot put.

9. Figure 2 is quite blurry and the font size is small.

10. The spacing between paragraphs is inconsistent.

11. References section lacks a more carefully compiled list, e.g. lines 527, 534, 554.

Reviewer #2: This manuscript examines the factors affecting talent development and athletic performance among Saudi Arabia's Twice-Exceptional Athletes, focusing on both internal psychological factors and external environmental influences. The study is underpinned by qualitative interviews and explores how these factors contribute to success at regional and international Paralympic competitions across various sports and regions within the Kingdom.

The selection of the sample and variables could be enhanced by including education level and profession as indicators of financial support and self-funding. This addition is particularly relevant given the disparities in financial backing and media coverage between Paralympic sports and other athletic disciplines. Moreover, the paper would benefit from a deeper contextualization within the current sports policy framework in Saudi Arabia, especially concerning disability and Paralympic sports. This should encompass aspects like financial support, dual careers, and access to facilities.

The discussion on regional discrepancies in success factors for the studied population is somewhat generic. A more detailed examination of specific cities that provide optimal conditions for these athletes would add value. Furthermore, the gender dimension warrants more comprehensive analysis and discussion.

There are also several stylistic concerns that require the authors' attention:

Ensure font consistency throughout the document. For example, the font in Table 2 differs from the rest of the paper.

Standardize table formatting to enhance readability, applying the style used in Table 1 to all subsequent tables.

Correct typographical errors, such as the one on line 367; it should read "this factor" instead of "his factor

6. PLOS authors have the option to publish the peer review history of their article (what does this mean?). If published, this will include your full peer review and any attached files.

Reviewer #1: No

Reviewer #2: No

---

## [Author Response · Author response to Decision Letter 0]

11 Sep 2024

i have answered all your comments in the file named Response to Reviewers

---

## [Decision Letter · Decision Letter 1]

14 Oct 2024

PONE-D-24-17887R1

The internal and external factors influencing Talent Development of athletic talent Among Saudi Arabia's Twice Exceptional Elite Athletes: A Comprehensive Study

PLOS ONE

Dear Dr.  Hassan,

Thank you for submitting your manuscript to PLOS ONE. After careful consideration, we feel that it has merit but does not fully meet PLOS ONE’s publication criteria as it currently stands. Therefore, we invite you to submit a revised version of the manuscript that addresses the points raised during the review process.

We look forward to receiving your revised manuscript.

Kind regards,

Tien-Chin Tan, Ph.D.

Academic Editor

PLOS ONE

Additional Editor Comments:

According to the reviewers, this manuscript may contribute to the literature to some extent. Nonetheless, the reviewers expressed reservations about the quality of this manuscript in its current form, particularly with regard to the methodology. There is still work to be done before the manuscript can be considered publishable. Our suggestion is that the author(s) heed the reviewers' comments and revise the manuscript accordingly as a result.

Reviewers' comments:

Reviewer's Responses to Questions

**Comments to the Author**

1. If the authors have adequately addressed your comments raised in a previous round of review and you feel that this manuscript is now acceptable for publication, you may indicate that here to bypass the “Comments to the Author” section, enter your conflict of interest statement in the “Confidential to Editor” section, and submit your "Accept" recommendation.

Reviewer #1: (No Response)

Reviewer #3: (No Response)

2. Is the manuscript technically sound, and do the data support the conclusions?

Reviewer #1: Partly

Reviewer #3: No

3. Has the statistical analysis been performed appropriately and rigorously? 

Reviewer #1: N/A

Reviewer #3: I Don't Know

4. Have the authors made all data underlying the findings in their manuscript fully available?

Reviewer #1: Yes

Reviewer #3: No

5. Is the manuscript presented in an intelligible fashion and written in standard English?

Reviewer #1: No

Reviewer #3: Yes

6. Review Comments to the Author

Reviewer #1: Greetings, authors,

Thank you for modifying and responding to our suggestions. Overall, the quality of the current study has improved. However, there are still a few points to consider and clarify. Here are some suggestions to consider before submitting the article to a journal.

Materials and Methods

1. Although the authors responded, "All requirements have been added to Table (1)", when reviewing Table 1, we cannot identify the respondent code, the respondent's disability classification, and the sport he/she plays. As an aside, the document currently has two Tables 1.

2. The author responds by saying, "The following passage has been added to clarify the validity of the interviews and qualitative data analysis.". It is in line with our recommendation. A review of the authors' manuscript, however, does not indicate that it has been added.

3. Also, although the authors responded to our suggestions, they did not include this information in their section on data collection.

Results

1. In line 206, the authors mention “Identifying the Identity of Twice-Exceptional Elite Athletes: A Mega Mode Approach”, but then go on to say that “This framework categorizes these factors into internal and external components, both of which are integral to motivating individuals toward achieving excellence”. In psychology, identity refers to traits, beliefs, personalities, appearances, and/or expressions that define a person or group. Therefore, it should be an internal component. In this context, it might be more appropriate for the author to use the term "influencing factors" rather than "identity."

2. In table 2, there are still no changes that would respond to our previous suggestions regarding the presentation of "The comparative analysis of external factors".

3. Only one change has been made to the separation of the findings from the discussion of previous studies. It is necessary to make many other adjustments, such as in lines 264-266.

Problems with Article Format

Throughout the article, there are several formal errors that need to be corrected by the author. The following is just a list of cases, and authors are encouraged to check each one throughout the text.

1. There is no space before [13] on line 52.

2. There is a need to clarify line 91.

3. There is a difference in font size between line 203 "Results" and other headings of the same level.

4. There is a need to adjust the distance between the lines on lines 318-319.

5. There is an inconsistency in the font size for Line 320.

6. The following lines are garbled: 523, 652, 717, 726.

Reviewer #3: I have the impression that the author(s) pointed out that interview data were collected from 21 respondents on 1 January 2023. And 26 hours of documentation......It is crucial to convince the reviewers that your data colletion process is scientific and trustful. Moreover, the author(s) need to re-frame the method section:

• who was to be studied and why;

• how respondents were recruited;

• the semi-structured interviewing approach;

• the number of people interviewed and the context in which the interviews took place;

• the approach to analysing the interview transcripts (thematic analysis).

7. PLOS authors have the option to publish the peer review history of their article (what does this mean?). If published, this will include your full peer review and any attached files.

Reviewer #1: No

Reviewer #3: No

---

## [Author Response · Author response to Decision Letter 1]

16 Oct 2024

Dear Dr. Tien-Chin Tan,

Thank you once again for your thoughtful and constructive feedback on our manuscript. We sincerely appreciate the time and effort you have dedicated to reviewing our work.

In response to your comments from the second round of reviews, we have made significant revisions to the manuscript to address your concerns and suggestions. Your insights have been invaluable in guiding us to improve the clarity, coherence, and overall quality of the research.

In this table you will find all the required responses and modifications.

Respond to all comments received via email

No. Comments Response

Materials and Methods

1 Although the authors responded, "All requirements have been added to Table (1)", when reviewing Table 1, we cannot identify the respondent code, the respondent's disability classification, and the sport he/she plays. As an aside, the document currently has two Tables 1. A new table was added with the number Table 1, which describes the participants according to the type of sport, type of disability, age, and timing of disability. Each participant was mentioned by his code, and the title of the table was Participant Distribution by Sport Type, Disability Type, Age, and Timing of Disability.

2 The author responds by saying, "The following passage has been added to clarify the validity of the interviews and qualitative data analysis.". It is in line with our recommendation. A review of the authors' manuscript, however, does not indicate that it has been added. The response to what the reviewers requested has been referred to and added to the manuscript.

3 3. Also, although the authors responded to our suggestions, they did not include this information in their section on data collection. Don, 

we added 

The following passage has been added to clarify the credibility of the interviews and qualitative data analysis:

Data from the participants were collected using the study tool through in-depth interviews with the athletes or their guardians, depending on the individual's situation. Two members of the research team (Fahad and Rifa) conducted the interviews, along with a former athlete with a disability who is also the Executive Director of a disability sports club. This individual holds a Master’s degree in Special Education and has 28 years of experience in the field. The data analysis process included several stages:

1. The researchers transcribed the interview and focus group recordings using Sonix (https://sonix.ai/), a platform designed for this purpose that supports Arabic and employs artificial intelligence for language and text recognition across contexts.

2. An exploratory content-driven analysis approach was used, which involves open coding and themes derived from the data, applicable to both probabilistic and non-probabilistic samples[49].

3. During the qualitative data analysis, the researchers performed triangulation of raw analysis by reviewing the data analysis process with a peer—a colleague in a gifted education PhD program—and a special education expert with 28 years of practical experience, known as peer review (Johnson & Christensen, 2019), to ensure the credibility of the analysis and review the main and sub-themes [48].

4. Additionally, the researchers presented the final analysis results to an independent researcher for external audit, a method involving external experts to assess the study's quality. The researcher reviewed the primary analysis and the initial review to determine the level of agreement and ensure the quality of identifying the main and sub-themes[48].

Results

1 In line 206, the authors mention “Identifying the Identity of Twice-Exceptional Elite Athletes: A Mega Mode Approach”, but then go on to say that “This framework categorizes these factors into internal and external components, both of which are integral to motivating individuals toward achieving excellence”. In psychology, identity refers to traits, beliefs, personalities, appearances, and/or expressions that define a person or group. Therefore, it should be an internal component. In this context, it might be more appropriate for the author to use the term "influencing factors" rather than "identity." The term has been changed and modified in the manuscript on line 252 to influencing factors//// instead of Identity.

2 In table 2, there are still no changes that would respond to our previous suggestions regarding the presentation of "The comparative analysis of external factors". We have made significant changes to address suggestions for a “comparative analysis of external factors.” A comprehensive comparison of various environmental factors has been included, including academic and economic levels, geographic location, timing of injury, talent discovery, and intensive training. The impact of each of these factors, both positive and negative, on athletes’ performance is illustrated, reflecting the impact of the surrounding environment on their success in sport. With this amendment, we hope that the information presented in Table 4 will be clearer and more comprehensive, meeting the reviewer’s expectations for an effective comparative analysis.

3 Only one change has been made to the separation of the findings from the discussion of previous studies. It is necessary to make many other adjustments, such as in lines 264-266. All modifications have been made and added.

Problems with Article Format

 There is no space before [13] on line 52. Done

 There is a need to clarify line 91. Done

 There is a difference in font size between line 203 "Results" and other headings of the same level. Done

 There is a need to adjust the distance between the lines on lines 318-319. Done

 There is an inconsistency in the font size for Line 320. Done

 The following lines are garbled: 523, 652, 717, 726. Done

Acknowledgments: In The authors extend their appreciation to the King Salman center For Disability Research for funding this work through Research Group no KSRG-2022-090. The funders had no role in study design, data collection and analysis, decision to publish, or preparation of the manuscript.

---

## [Decision Letter · Decision Letter 2]

21 Nov 2024

PONE-D-24-17887R2The internal and external factors influencing Talent Development of athletic talent Among Saudi Arabia's Twice Exceptional Elite Athletes: A Comprehensive StudyPLOS ONE

Dear Dr. Hassan,

Thank you for submitting your manuscript to PLOS ONE. After careful consideration, we feel that it has merit but does not fully meet PLOS ONE’s publication criteria as it currently stands. Therefore, we invite you to submit a revised version of the manuscript that addresses the points raised during the review process.

We look forward to receiving your revised manuscript.

Kind regards,

Tien-Chin Tan, Ph.D.

Academic Editor

PLOS ONE

Journal Requirements:

Additional Editor Comments:

Thanks for revising the original manuscript. In the revised manuscript, there have been improvements. Although enhancements have been made, reviewers remain concerned about certain issues. Please make an adjustment or provide clarification to clarify the reviewers' concerns. The manuscript can be accepted after a few minor revisions. Looking forward to receiving the further revised version.

Reviewers' comments:

Reviewer's Responses to Questions

**Comments to the Author**

1. If the authors have adequately addressed your comments raised in a previous round of review and you feel that this manuscript is now acceptable for publication, you may indicate that here to bypass the “Comments to the Author” section, enter your conflict of interest statement in the “Confidential to Editor” section, and submit your "Accept" recommendation.

Reviewer #1: All comments have been addressed

Reviewer #3: All comments have been addressed

2. Is the manuscript technically sound, and do the data support the conclusions?

Reviewer #1: Yes

Reviewer #3: Yes

3. Has the statistical analysis been performed appropriately and rigorously? 

Reviewer #1: N/A

Reviewer #3: Yes

4. Have the authors made all data underlying the findings in their manuscript fully available?

Reviewer #1: Yes

Reviewer #3: Yes

5. Is the manuscript presented in an intelligible fashion and written in standard English?

Reviewer #1: Yes

Reviewer #3: Yes

6. Review Comments to the Author

Reviewer #1: Dear authors,

Thank you for accepting our change suggestions and promptly submitting a corrected version. However, after re-examining the revised article, there are still some areas to be corrected.

Materials and methods

1. The authors have added Table 1 to the revised version of the article to show the Participant Distribution. This allows readers to understand the respondents' attributes. However, in general, the authors will point out the location of the table and explain the meaning of the table in the text.

Results

1. Lines 378-403, both paragraphs are identical.

Discussion

1. lines 542-544, there is a problem with the subparagraphs. 2.

2. Suggest that the author use a topic-based presentation in the discussion section to make it easier for the reader to understand what the author is trying to discuss.

Problems with the Article Format

1. On line 119, citation formatting should be standardized. In every section of the article, fonts should be used.

2. It would be helpful if you standardized the font and font size throughout the article.

3. On lines 538 and 546, please delete the extra parentheses.

Reviewer #3: All of my concerns surrounding the data collection and analysis have been solved. Well done! Now, I would say that it is a nice qualitative research paper.

7. PLOS authors have the option to publish the peer review history of their article (what does this mean?). If published, this will include your full peer review and any attached files.

Reviewer #1: No

Reviewer #3: No

---

## [Author Response · Author response to Decision Letter 2]

22 Nov 2024

Dear Dr. Tien-Chin Tan,

Thank you once again for your thoughtful and constructive feedback on our manuscript. We sincerely appreciate the time and effort you have dedicated to reviewing our work.

In response to your comments from the third round of reviews, we have made significant revisions to the manuscript to address your concerns and suggestions. Your insights have been invaluable in guiding us to improve the clarity, coherence, and overall quality of the research.

In the attached table, you will find detailed responses to all comments received via email, along with the corresponding modifications made to the manuscript.

Additionally, we would like to kindly inform you that our research team is hopeful for an expedited review process. This will enable us to submit the manuscript to the funding agency before the previously mentioned deadline of 30 November, ensuring we can present it to the King Salman Center for Disability Research in a timely manner.

Thank you for your continued support and understanding.

No. Comments Response

Journal Requirements:

1 Please review your reference list to ensure that it is complete and correct. The reference list has been thoroughly reviewed to ensure it is complete and accurate. We also use the EndNote application for writing and managing references.

2 If you have cited papers that have been retracted, please include the rationale for doing so in the manuscript text, or remove these references and replace them with relevant current references. Thank you for your note regarding retracted papers. I would like to confirm that I have thoroughly reviewed the references cited in my manuscript, and none of them include retracted papers.

3 Any changes to the reference list should be mentioned in the rebuttal letter that accompanies your revised manuscript. References 77 to 83 have been updated in the revised reference list.

References 78 to 77 have been updated in the revised reference list.

References 81 to 78 have been updated in the revised reference list.

References 82 to 79 have been updated in the revised reference list.

References 80 to 85 have been updated in the revised reference list.

References 83 to 80 have been updated in the revised reference list.

References 84 to 81 have been updated in the revised reference list.

EndNote automatically updates the reference list in the manuscript body and reference list.

4 If you need to cite a retracted article, indicate the article’s retracted status in the References list and also include a citation and full reference for the retraction notice. I confirm that my manuscript does not include citations from retracted articles. All references have been carefully reviewed to ensure their validity and relevance.

Additional Editor Comments:

1 Thanks for revising the original manuscript. In the revised manuscript, there have been improvements. Although enhancements have been made, reviewers remain concerned about certain issues. Thank you for your positive feedback on the revisions made to the manuscript. We are glad that the improvements have been noted, and we appreciate your acknowledgment of the changes. We will continue to work on refining the manuscript further to address any remaining concerns.

2 Please make an adjustment or provide clarification to clarify the reviewers' concerns. The manuscript can be accepted after a few minor revisions. Looking forward to receiving the further revised version. Thank you for your feedback and for recognizing the improvements made in the revised manuscript. We appreciate the reviewers' insightful comments and understand that some concerns remain. We are currently working on addressing these points by making the necessary adjustments and providing clarifications to ensure the manuscript fully meets the required standards. We will carefully review the reviewers' concerns and make the appropriate revisions to resolve any remaining issues. Thank you again for your valuable feedback, and we hope that these amendments will be comprehensive, provide all clarifications, and remove all concerns.

Comments to the Author

 Materials and methods

1 The authors have added Table 1 to the revised version of the article to show the Participant Distribution. This allows readers to understand the respondents' attributes. However, in general, the authors will point out the location of the table and explain the meaning of the table in the text. We appreciate your valuable comment. In response, we have included a clear reference to Table 1 in the appropriate section of the manuscript, immediately following the table. Additionally, we have provided a detailed explanation of its content within the text to help readers better understand the participant distribution and the significance of the data presented in the table. The explanation is as follows:

Table 1 highlights the distribution of volunteer participants in the study based on sport type, disability type, age, and disability onset. The key findings from Table 1 can be summarized as follows:

Participant Diversity (Age and Disability Type):

Table 1 reveals significant diversity in age groups, with participants ranging from 19 years old (the youngest) to 56 years old (the oldest).

There is also notable variation in disability types, including physical impairments such as paraplegia and limb amputations, as well as intellectual disabilities and autism spectrum disorders.

Disability Onset and Causes:

Disabilities are categorized into those present since birth or early childhood (e.g., polio or oxygen deprivation during birth) and those acquired later due to accidents or medical errors (e.g., injuries caused by accidents or vaccination complications).

The timing of disability onset reflects its psychological and social impact, with some participants acquiring their disabilities during critical life stages, such as adolescence or adulthood.

Sport Type:

Participants engage in a wide range of sports that require diverse physical abilities:

Individual sports: such as powerlifting, track and field (e.g., shot put, wheelchair racing).

Team sports: such as wheelchair basketball.

Para-specific competitive sports: including Boccia and classification-based sports (e.g., T and F categories).

This diversity demonstrates the opportunities available for individuals with disabilities to participate in various sports, taking into account the specific classifications associated with each disability type.

Results

1 Lines 378-403, both paragraphs are identical. Thank you for pointing that out. In response, we have removed the duplicated content between lines 378-403 to ensure clarity and avoid repetition.

Discussion

1 lines 542-544, there is a problem with the subparagraphs. 2. Thank you for pointing out the problem in lines 542 to 544 regarding the subparagraphs. I have reviewed the content and made the necessary adjustments to ensure clarity and proper formatting. This was done by modifying and arranging the paragraphs of the entire discussion and placing headings so that the reader can easily browse the discussion

2 Suggest that the author use a topic-based presentation in the discussion section to make it easier for the reader to understand what the author is trying to discuss. Numerous studies in the literature have affirmed the potential for intentional development across various domains [76]. However, a pertinent question arises: What were the key factors responsible for nurturing sports talent among elite, twice-exceptional athletes?. 

To elucidate these capacities and comprehend their operational mechanisms, we scrutinized the experiences of individuals who had attained the benchmarks indicative of talent. Accordingly, we investigated both internal factors and environmental factors, discerning how they catalyzed talent development. Given the diverse experiences of the participants within the twice-exceptional category, we conducted comparative analyses to benchmark their actions' efficacy and their influence on developmental trajectories under varying circumstances, contrasting ideal conditions with those lacking such support. 

The participants' experiences unveiled a synergistic interplay among internal factors and other components, alongside interactions with environmental factors, both individually and collectively. Furthermore, a positive and catalytic interaction was observed between internal and environmental factors. Each factor stimulated the emergence and evolution of the other, contingent upon the individual's age and developmental stage within the talent trajectory. 

 The role of internal factors in developing sports performance:

The transition into the realm of sports followed a similar trajectory for participants, yet the nuanced interplay of various capabilities, representing the intended practice method, differed significantly. This variance influenced the breadth and depth of skill development achievement. For many participants, adolescence marked the on-set of their sports journey, coinciding with the commencement of diverse sporting pursuits[51]. These endeavors often catalyzed emotional qualities such as a love for challenge and self-assertion [76]. 

Motivation serves as an internal engine for mastering performance and improving psychological, social, and life skills [77]. Previous research emphasizes the significance of internal emotional components such as passion, satisfaction, desire, and self-assertion in enhancing athletic performance within structured practice environments[63]. Passion and strong emotions, sometimes referred to as "hallucinations," have varied effects on athletes. They not only contribute to feelings of achievement and vitality during moments of victory but also catalyze athletes to set objectives and adopt individual methodologies, ultimately leading to excellence[64]. 

Studies centered on the hypotheses underlying the model and the requirements of the twice-exceptional population emphasized the importance of leveraging strengths, addressing weaknesses, implementing credible interventions, and tailoring approaches to individual needs[78-81]. These efforts were aimed at fostering supportive internal psychological traits, such as passion, perseverance, and assertiveness, which were crucial for talent development. 

Notably, many individuals struggled to accept their new circumstances, which hindered their participation in sports activities. This finding is consistent with research by Boyce and Wood (2011), which suggests that adapting to a disability is easier when it occurs earlier in life. In contrast, those who adapted successfully achieved remarkable accomplishments and competed at various levels, balancing their athletic pursuits with other aspects of life.[73, 82]. Moreover, Gignac et al. (2000) observed that individuals with chronic disabilities often develop effective coping mechanisms, further emphasizing the importance of psychological resilience in overcoming challenges[73]. These insights highlight the importance of early adaptation to disability and the role of active compensatory strategies in achieving success in sports and maintaining a balanced life. 

Motivation surged when favorable environmental factors were aligned with internal traits, enabling disabled individuals to achieve remarkable performance commensurate with their age and athletic trajectory. These internal components served as catalysts for overcoming challenges and sustaining motivation and growth.

Our findings indicate that elite athletes navigate this transition from capability to competence more easily than from competence to experience. Internal components play a crucial role in this process, enabling athletes to enhance their performance and surpass competitors. While environmental factors like academic and economic levels, as well as geographical location, have less influence, internal motivation for achievement and self- assertion emerges as key drivers, often described as falling in love [19].

The role of external factors in developing sports performance:

Disparities in parental education and economic status, along with opportunities for talent discovery linked to geographical location, significantly influenced the development of emotional qualities essential for nurturing sports talent. Champions who benefited from supportive environments and optimal interactions achieved exceptional performance levels, earning numerous international gold medals and setting records. In contrast, those lacking such nurturing opportunities did not attain comparable levels of skill and accomplishment. While interaction dynamics played a crucial role, the presence of abilities and the timing of their development also impacted outcomes. Geographical suitability, coupled with parental support and educational advantages in early life, facilitated the athletic development of twice-exceptional individuals, enhancing their chances of achieving high levels of success. According to the "Mega Model," timely opportunities were critical for sustained growth, as each domain required specific timing for optimal development [51]. It is evident that capabilities, when coupled with a systematic arrangement of quality, size, and timing, facilitate a controlled interaction process with a positive impact on skill development and achievement levels.

The model suggests that capabilities undergo a developmental transition toward competence, leading individuals to become experienced and eventually distinguished in their field[83]. 

This stage demanded that elite athletes possess psychological and social skills essential for navigating challenges, asserting themselves, confronting obstacles, and outperforming competitors. These skills are often cultivated through immersion in the field and gaining experience through sports practices, as evidenced by elite athletes' experiences in the Olympic Games for the disabled. Research suggests that individuals with more experience tend to develop psychological and social skills such as leadership and communication[84]. Additionally, elite athletes needed to master the intricacies of professionalism and performance arts, underscoring the importance of having expert coaches with a high level of professionalism to facilitate seamless performance and successful navigation of challenges [85]. Participants highlighted the significance of boot camps and the role of professional trainers in accelerating their transition towards excellence and expertise in the field.

 We observed the divergence between internal and external factors across various stages of development and noted how the timing of these components influenced talent development trajectories, particularly in the case of twice-exceptional individuals. Timing emerged as a fundamental aspect of the Mega Model theory, wherein the presence of components at opportune moments significantly impacted talent development outcomes. Additionally, each domain had unique temporal needs, highlighting the importance of considering timing in talent development strategies.

Problems with the Article Format

1 On line 119, citation formatting should be standardized. In every section of the article, fonts should be used. Done

As noted by Moser and Korstjens, face-to-face in-depth interviews are the most propitious for phenomenological research methodology. Similarly, Creswell and Poth emphasized this method [40, 45].

2 It would be helpful if you standardized the font and font size throughout the article. Done

3 On lines 538 and 546, please delete the extra parentheses. Done

---

## [Decision Letter · Decision Letter 3]

6 Dec 2024

The internal and external factors influencing Talent Development of athletic talent Among Saudi Arabia's Twice Exceptional Elite Athletes: A Comprehensive Study

PONE-D-24-17887R3

Dear Dr. Mohamed Dahy Hassan,

We’re pleased to inform you that your manuscript has been judged scientifically suitable for publication and will be formally accepted for publication once it meets all outstanding technical requirements.

Kind regards,

Tien-Chin Tan, Ph.D.

Academic Editor

PLOS ONE

Additional Editor Comments (optional):

I would like to congratulate the authors for the diligent work they put into revising this manuscript. All comments and suggestions received during the revision process have been successfully incorporated. Congratulations.

Reviewers' comments:

Reviewer's Responses to Questions

**Comments to the Author**

1. If the authors have adequately addressed your comments raised in a previous round of review and you feel that this manuscript is now acceptable for publication, you may indicate that here to bypass the “Comments to the Author” section, enter your conflict of interest statement in the “Confidential to Editor” section, and submit your "Accept" recommendation.

Reviewer #1: All comments have been addressed

2. Is the manuscript technically sound, and do the data support the conclusions?

Reviewer #1: Yes

3. Has the statistical analysis been performed appropriately and rigorously? 

Reviewer #1: Yes

4. Have the authors made all data underlying the findings in their manuscript fully available?

Reviewer #1: Yes

5. Is the manuscript presented in an intelligible fashion and written in standard English?

Reviewer #1: Yes

6. Review Comments to the Author

Reviewer #1: Dear author, greetings.

Thank you to the author group for accepting our suggestions and correcting the manuscript. The manuscript still has a few formatting errors, but they did not interfere with the reading fluency. Additionally, the overall content has been clarified. Therefore, we do not have any additional revision suggestions. My sincere congratulations.

7. PLOS authors have the option to publish the peer review history of their article (what does this mean?). If published, this will include your full peer review and any attached files.

Reviewer #1: No

---

## [Editor Report · Acceptance letter]

12 Dec 2024

PONE-D-24-17887R3 

PLOS ONE

Dear Dr. Hassan, 

I'm pleased to inform you that your manuscript has been deemed suitable for publication in PLOS ONE. Congratulations! Your manuscript is now being handed over to our production team.

Kind regards, 

on behalf of

Professor Tien-Chin Tan 

Academic Editor

PLOS ONE